# Plant DNA Methylation: An Epigenetic Mark in Development, Environmental Interactions, and Evolution

**DOI:** 10.3390/ijms23158299

**Published:** 2022-07-27

**Authors:** Francesca Lucibelli, Maria Carmen Valoroso, Serena Aceto

**Affiliations:** 1Department of Biology, University of Naples Federico II, 80126 Naples, Italy; 2Department of Agricultural Sciences, University of Naples Federico II, 80055 Portici, Italy; mariacarmen.valoroso@unina.it

**Keywords:** DNA methylation, plant epigenetics, gene expression, plant genomic imprinting, environmental adaptations

## Abstract

DNA methylation is an epigenetic modification of the genome involved in the regulation of gene expression and modulation of chromatin structure. Plant genomes are widely methylated, and the methylation generally occurs on the cytosine bases through the activity of specific enzymes called DNA methyltransferases. On the other hand, methylated DNA can also undergo demethylation through the action of demethylases. The methylation landscape is finely tuned and assumes a pivotal role in plant development and evolution. This review illustrates different molecular aspects of DNA methylation and some plant physiological processes influenced by this epigenetic modification in model species, crops, and ornamental plants such as orchids. In addition, this review aims to describe the relationship between the changes in plant DNA methylation levels and the response to biotic and abiotic stress. Finally, we discuss the possible evolutionary implications and biotechnological applications of DNA methylation.

## 1. Introduction

Epigenetic modifications are heritable chemical changes that do not alter the DNA nucleotide sequence but can influence the phenotype [1,2]. Epigenetics is a fascinating topic in plant biology because epigenome dynamism and plasticity affect plant development and evolution in response to the environment.

The epigenetic marks include chemical modifications of the DNA (i.e., methylation) and numerous post-translational modifications of histone proteins, such as acetylation, methylation, ubiquitination, sumoylation, and phosphorylation. These changes generally occur at the histone N-terminal tails [3,4]. In addition to covalent modifications of DNA and histones, epigenetic mechanisms include the activity of non-coding RNA molecules [5,6,7] and chromatin remodeling factors that control the nucleosome position, assembly, and disassembly [8,9]. These modifications can alter the accessibility of the genomic regions and regulate gene expression, resulting in phenotypic plasticity. The consequent ability to rapidly modulate the responses to environmental changes is crucial for all living organisms, especially for sessile such as plants [10].

DNA methylation is one of the most studied epigenetic mechanisms [10]. Plant DNA methylation occurs through the link of a methyl group (-CH3) at cytosine in symmetric, CG and CHG; and asymmetric, CHH, contexts (where H is any nucleotide except G), carried out by a family of enzymes called DNA methyltransferases [3,11,12] (Figure 1).

In the thale cress (*Arabidopsis thaliana*), a model species for plant genetics, the CG DNA methylation is maintained by the METHYLTRANSFERASE 1 (MET1) during DNA replication [9], while the CHROMOMETHYLASE 2 and 3 (CMT2 and CMT3) generally methylate the CHG and CHH contexts. In particular, CMT2 acts mainly in the CHH context and CMT3 acts in the CHG context [11,13,14]. In contrast, the DNA methyltransferase DOMAINS REARRANGED METHYLTRANSFERASE 2 (DRM2) is involved in maintaining the existing methylation landscape in both symmetric and asymmetric contexts, and it is also responsible for de novo DNA methylations. DRM2 is guided to the target sequence by small RNA molecules in a plant-specific pathway known as RNA-directed DNA methylation (RdDM) [3,11,15,16].

In *Arabidopsis*, RdDM can occur in two different ways: canonical and non-canonical. The canonical pathway is mainly involved in maintaining methylation in heterochromatic regions. During canonical RdDM, the RNA Polymerase IV (Pol IV) interacts with the chromatin remodeler CLASSY 1 (CLSY1) and its interactor SAWADEE HOMEODOMAIN HOMOLOG 1 (SHH1). This complex binds to heterochromatin, and RNA Pol IV transcribes short single-strand RNAs (ssRNAs), approximately 30–45 nucleotides in length. The RNA-directed RNA polymerase RDR2 and RNA Pol IV convert these ssRNAs into double-strand RNAs (dsRNAs). Then, the endoribonuclease DICER-LIKE 3 (DCL3) cleaves the dsRNAs into 24 nucleotides sRNAs. The ARGONAUTE proteins (AGO4, AGO6) convey single strands of the sRNAs toward the complementary RNA transcribed by the plant-specific RNA Polymerase V, recruiting the methyltransferase DRM2 that methylates the neighboring DNA.

The non-canonical pathway is less frequent and includes many variations of the canonical pathway, generally resulting in de novo DNA methylation (for example, methylation of newly transposed mobile elements). In the non-canonical pathway, the biogenesis of sRNAs is not only mediated by Pol IV—RDR2—DCL3, but they can be originated by many other mechanisms such as RNA Polymerase II transcription [17,18,19,20,21,22].

Methylation levels are finely controlled, and the dynamic of DNA methylation/demethylation is a rigorously orchestrated process. In plants, demethylation can be a passive mechanism with the loss of the methylation signal during DNA replication, or an active mechanism mediated by specific enzymes [23] through the DNA base excision repair (BER) [24,25]. Specific plant DNA glycosylases recognize methylated cytosine in any sequence context and break the bond with the deoxyribose sugar, originating an abasic site that will be repaired by DNA polymerase and ligase.

The four DNA demethylases identified in *Arabidopsis* are the REPRESSOR OF SILENCING 1 (ROS1), DEMETER (DME), and DEMETER-LIKE PROTEIN 2 and 3 (DML2, DML3) [26,27,28,29,30,31] (Figure 1). In mammals, the Ten-Eleven Translocation methylcytosine dioxygenase (TET) enzymes play an important role in demethylation, catalyzing the oxidation of the methylated cytosine [32]. However, plant TET-like enzymes are not yet characterized, even if the existence of the oxidative product during active demethylation suggests the possible involvement of TET-like proteins in this process [33]. Moreover, the overexpression of a human TET protein in *Arabidopsis* induces significant genome demethylation, further supporting the existence of TET-like proteins in plants [34].

In addition to cytosine methylation, generally associated with transcriptional repression, the plant genome can be dynamically methylated on N^6^-adenine. The 6-methyl-adenosine (6mA) is associated with expressed genes and seems to be very receptive to environmental stimuli [35,36].

This review will focus on the molecular and cellular functions of DNA methylation in plants, highlighting its role in the plant’s life cycle, response to environmental changes, and evolution.

## 2. Role of DNA Methylation in Plant Cells

The level of DNA methylation deeply influences the plant genome structure and activity. Numerous and different molecular mechanisms are affected by the action of DNA methylation, underlining the importance of this epigenetic mechanism in the plant cell.

### 2.1. Gene Expression

A critical molecular consequence of DNA methylation is the regulation of gene expression (Figure 2A) [17].

Usually, due to the promoter region methylation, the binding of transcription activators is prevented, and that of transcription repressors is improved, leading to the inactivation or reduction of transcription. Moreover, gene expression regulation results from the cooperation of different epigenetic mechanisms that influence each other, for example, DNA methylation and histone modifications. The N-terminal tails of histones can undergo numerous modifications promoted by specific enzymes. Some of these modifications, such as histone acetylation or H3 lysine 4 monomethylation (H3K4me1), relax the chromatin, thus facilitating the transcription. Generally, regions with a low methylation level also present other modifications that promote transcription, e.g., histone acetylation. However, in some cases, poorly methylated and actively transcribed regions can show histone hypo-acetylation, suggesting that sometimes the loss of the DNA methylation is a mechanism that, alone, can promote the activation of transcription [17,37,38].

Less frequently, DNA methylation can also activate gene transcription, even if the molecular mechanism involved is not yet understood [17].

In addition to the promoter, methylation can also occur within the full-length coding region, generally in the central exons rather than in the transcription start and end regions [17,39]. Unlike promoter methylation, which suppresses or decreases the transcription of the downstream gene, methylation in the coding region inhibits transcript elongation. In this way, the transcription occurs from the transcription start site to the methylation site, with the production of a smaller, possibly inactive transcript [40].

Intragenic methylation plays a relevant regulatory and biological role, preventing the activation of cryptic transcription start sites [36]. In the plant meristem cells, aberrant transcription produces double-stranded RNAs that trigger a feedback methylation mechanism via RdDM and repress the cryptic promoters from which they originated. The methylation of these cryptic transcription start sites is maintained in the differentiated tissues where the chromatin preserves its inaccessible structure, and the aberrant transcription is suppressed [40,41] (Figure 2B).

### 2.2. Transposon Mobility

DNA methylation is involved in maintaining plant genome stability by regulating transposon mobility. The methylation of the transposable elements occurs in all sequence contexts and in the genome of all plant species, from mosses to angiosperms (Figure 3) [42,43,44,45,46,47]. In particular, the RdDM pathway maintains the CHH sequence methylation in short and large transposon ends, whereas CMT2 methylates the internal regions of large transposons [14,48].

The transposon activity within the genome can disrupt the structure of a gene or a regulatory region. Therefore, methylation acts as a cellular defense mechanism by reducing the transposition rate of the mobile elements [17,42] (Figure 2C). Transposon silencing can also cause the spread of methylation to the flanking areas, thus generating differentially methylated regions [49,50]. In the plant genomes, such as maize (*Zea mays*), the hypermethylated CHH islands maintained by the RdDM mechanism are fundamental to separating the transcriptionally active regions and inactive transposons. In this way, the eventual activation of a silenced transposable element by the positional effect of nearby active euchromatin is unlikely [17,51].

Although its critical function in genome stability, sometimes DNA methylation alone is not sufficient to ensure the silencing of the transposable elements. As mentioned above, additional marks, like histone modifications, are necessary to regulate transposon mobility. For example, dimethylation of H3K9 (H3K9m2) is a significant mark in heterochromatin formation. In particular, in the *Arabidopsis* genome, there is a correlation between H3K9m2 levels and transposons present in euchromatic chromosome arms, suggesting this modification has a role in the silencing of transposons [52,53,54].

### 2.3. Chromosome Interactions

Another function of DNA methylation is the regulation of chromosome interactions. The hypermethylation of pericentromeric regions and some interactive heterochromatin islands (IHI) of euchromatic chromosome arms generate an interaction network (Figure 2D). In *Arabidopsis*, this nuclear complex is called KNOT and is present in all five chromosomes [55,56]. The KNOT complex has a relevant role in maintaining genome integrity, which is threatened by the jumping of transposons [48].

Genomic analyses of *Arabidopsis* have demonstrated the effect of DNA methylation on chromosome interactions, showing that the genomic regions with a high methylation level are less engaged in long-range chromatin contacts than the low methylated regions [57].

The control of chromosome interactions by DNA methylation is an important mechanism affecting gene expression. Promoters control gene expression in addition to some regulatory regions that can be very distant from the transcription start site (TSS). Although distant enhancers in the plant genomes are still unknown, this type of regulation cannot be excluded. Some speculative models describe how the control of chromatin landscape by methylation level could influence long-range chromosomal interactions and gene expression. In particular, inhibiting the formation of 3D spatial interactions between chromosomes prevents the regulation of gene expression through distant regulatory elements, such as enhancers. These models suggest that RdDM can prevent the binding of the architectural proteins blocking the formation of chromosomal interactions [57].

### 2.4. Biogenesis of circRNAs

Circular RNAs (circRNAs) are non-coding RNA molecules that have recently been characterized, and much is still unknown about their formation and role. They have been described in different angiosperm genomes such as *Arabidopsis* [58], rice (*Oryza sativa*) [59], and maize [60]. The linear mRNA molecules are generated from the canonical splicing mechanism where the upstream (5′) splicing donor site has linked to the downstream (3′) splicing acceptor site. In contrast, circRNAs are produced at the post-transcriptional level by a pre-mRNA back-splicing. The downstream splicing donor site is ligated with an upstream splicing acceptor site, and the 3′ and 5′ ends are covalently closed. circRNAs do not possess a polyadenylated tail and are more stable and resistant to exonuclease degradation than linear RNAs [61,62,63].

Interestingly, recent findings suggest that DNA methylation could also be involved in the biogenesis of circRNAs. For example, circRNA-seq data in the moso bamboo (*Phyllostachys edulis*) show a high methylation level in flanking intron regions of circRNAs (Figure 2E) [64]. Although the biological role of circRNAs is still unclear, they could be involved in miRNA binding and regulation of gene expression [65,66], generating a new level of epigenetic regulation through DNA methylation.

### 2.5. RNA Methylation

Methylation is a chemical modification that does not act only on DNA; in fact, it also plays a central role in mRNA stability and splicing, affects mRNA processing by generating alternative polyadenylation sites that cause the production of incomplete and unproductive transcripts, and influences the mRNA migration in the cytoplasm and its translation (Figure 2F) [67,68,69,70,71,72,73,74,75,76,77].

An analysis of plant epitranscriptome reveals that the most common post-transcriptional RNA modifications occur at N6-methyladenosine (m6A) and 5-methylcytidine (m5C) [78]. These two modifications are located in different regions of the transcripts. Near the stop codons and at 3′ untranslated regions (3′ UTR), m6A is abundant, while m5C is more frequent in coding sequences (CDS) [78].

The methylation level of the transcripts is dynamic and changes between tissues and different stages of plant development. This variation is due to the action of specific writer, eraser, and reader enzymes that catalyze epigenetic modifications. The writers generate the chemical modifications, the readers recognize and identify them, and the erasers remove them. The variation of these enzyme levels reflects the complexity of the epigenetic landscape and its dynamism [78,79,80,81,82,83,84].

In *Arabidopsis*, the m6A modifications on RNA depend on the specific complex that includes the methyltransferase A (MTA) that interacts with the methyltransferase B (MTB) and other factors such as FKBP12 interacting protein 37 (FIP37) [85,86]. The writers TRM4A and TRM4B perform the RNA methylation of m5C: the first is involved in the m5C methylation of tRNAs, and the second acts on mRNAs [87]. Among the erasers, thirteen enzymes have been identified in *Arabidopsis*. They belong to the ALKBH family and catalyze the removal of the m6A modifications [88]. Three YTHD proteins represent the m6A readers of *Arabidopsis* so far [89]. The expression of all these enzymes varies under abiotic stress conditions, showing the importance of the regulation of RNA chemical modifications. These enzymes can control the RNA molecule stability and fate by modifying specific nucleotides, thus affecting plant response to environmental conditions [84].

## 3. DNA Methylation during Plant Development

Variation of DNA methylation in different tissues and during the phases of plant development considerably affects plant physiology [17].

### 3.1. Genomic Imprinting

DNA methylation has a critical role in fertilization, the first step of the plant life cycle. During this stage, genomic imprinting can occur at specific loci through differential methylation of the two alleles dependent on their parental origin.

In angiosperms, female gametogenesis occurs in the ovary, where the meiosis of the megasporocyte produces four haploid megaspores. After the degeneration of three megaspores, the only surviving undergoes three sequential mitoses, generating a cell with eight nuclei. Two polar nuclei migrate at the cell center and form the central cell. Three nuclei move upwards, arising antipodal cells, and three migrate at the bottom, generating the two lateral synergid cells and the central egg cell. The pollinic granule is constituted by two spermatic cells and a vegetative cell. The latter forms the pollen tube that directs sperm cells in the ovary during fertilization. One of the two spermatic cells fuses with the egg cell forming the diploid zygote. The other sperm cell combines with the central cell, thus producing the triploid endosperm that nourishes the growing embryo [90]. This kind of megagametophyte is named *Polygonum*-type and characterizes more than 70% of angiosperm species, including *Arabidopsis* [91].

Genomic imprinting is a regulatory phenomenon involved principally in the embryo’s growth. It was first described in mammals, where the parental conflict theory explains the functional meaning of genomic imprinting [92]. Applying this theory to plants highlights the different parental contributions to the seed size [93,94]. The growth of the seed depends principally on the maternal food reserves, resulting in higher metabolic costs for maternal tissues than for paternal ones. For this reason, the maternal genes that control the seed size have to ensure a balance between an optimal size, the energy expense of the mother, and the energetic resources that other seeds already in the developing stage need for growth.

In contrast, the paternal tissues do not directly experience the metabolic expense for developing the seed. The only effect of the larger seed suffered by paternal tissue is indirect because there is a lower chance of reproduction due to the presence of fewer ovules to fertilize [92]. Consequently, the genes that repress endosperm growth and reduce the nutrient flow to the embryo are expressed by maternal alleles. On the contrary, the genes that increase endosperm development and stimulate the nutrient flow to the embryo are expressed by paternal alleles [95].

In the maternal central cell, an effective demethylation activity occurs before fertilization. Consequently, in the endosperm, maternal alleles are less methylated and more expressed than paternal ones [96,97,98]. Many imprinted genes, including transcription factors and protein involved in chromatin remodeling, have a significant regulative role in seed development and endosperm proliferation. For example, in the central cell of *Arabidopsis*, DME demethylates the genes *FLOWERING WAGENINGEN* (*FWA*), *MEDEA* (*MEA*), and *FERTILIZATION-INDEPENDENT SEED 2* (*FIS2*), while in the spermatic cell, these loci are methylated and silenced. In this way, after fertilization, only the *MEA*, *FIS2*, and *FWA* maternal alleles are expressed in the endosperm [99,100,101].

MEA is a Polycomb group protein (PcG) and is the homolog of Enhancer of zeste of *Drosophila* [100]. The PcG proteins of *Drosophila* have a role in maintaining the silenced state of the target genes through chromatin remodeling. The MEA protein of *Arabidopsis* suppresses central cell proliferation and endosperm development, regulating gene transcription by controlling chromatin accessibility. For this reason, the *MEA* maternal allele is expressed, as opposed to the paternal one [102]. As MEA, FIS2 is also a PcG protein, the homolog of the *Drosophila* Suppressor of zeste12 SU(Z)12. In *Arabidopsis*, FIS2 and MEA function in similar protein complexes playing a role in the transcriptional regulation of target genes [100,103].

In contrast to *MEA* and *FIS2*, involved in seed development, the imprinting of the *FWA* gene is fundamental for flowering. The ectopic expression of *FWA* and the alteration of the imprinting in *fwa* mutants are responsible for the plant’s late flowering [99].

In the central and vegetative cells of *Arabidopsis* [97], rice [104], and maize [105], DME can also demethylate mobile elements. This activates the transcription of small RNA molecules delivered to the egg and sperm cells, reinforcing transposon silencing in the gametes [97,106,107,108,109].

To date, genomic imprinting is described only in species with the *Polygonum*-type embryo sac, the most common type among angiosperms [110]. Future studies on genomic imprinting in taxa with other types of embryo sacs (e.g., *Allium*-, *Oenothera*-, *Adoxa*-type, etc.) will enhance our knowledge of the epigenetic mechanisms involved in angiosperm fertility.

### 3.2. Floral Pigmentation, Floral Scent, and Photosynthesis

Besides fertilization, epigenetic modifications are powerful regulators of many other biological processes in plants. For example, DNA methylation can influence the floral pigmentation in the *Oncidium* orchid.

The labellum of the *Oncidium* Gower Ramsey (GR) flower has an intense yellow coloration due to the accumulation of carotenoid pigments. The genetic pathway underpinning this pigmentation is known [111], and its variation is responsible for the white flower color in *Oncidium* White Jade (WJ). The different color patterns of these orchid cultivars depend on the differential expression of carotenoid-related genes in response to their methylation levels. In particular, in *Oncidium* GR, the *OgCCD1* gene is methylated and silenced. On the contrary, in WJ cultivar tissues, the lack of promoter methylation and the consequent active expression of the *OgCCD1* gene promotes carotenoid degradation resulting in a white flower [112].

Another characteristic of the pigmentation pattern of the *Oncidium* GR flower is the presence of a red streak in the perianth, absent in the cultivar *Oncidium* Honey Dollp (HD). In this case, the different coloration of the orchid floral organs depends on the methylation levels of the genes involved in the anthocyanin-biosynthetic pathway. The methylation of the *OgCHS* gene promoter in the HD cultivar is responsible for the absence of anthocyanin accumulation. In contrast, the unmethylated gene is effectively expressed in GR [113].

DNA methylation plays an active role in the flower color formation in many other ornamental plants. In the hall crabapple (*Malus halliana*), petal coloration changes from red to pale pink during development, and this variation is associated with the downregulation of many genes. In particular, the promoter of *MhMYB10*, a gene of the anthocyanin biosynthesis pathway, is highly methylated, resulting in a reduction of the *MhMYB10* gene expression and a decrease in anthocyanin accumulation [114].

The methylation of gene promoters predominantly causes gene expression variation during plant development and growth. The alteration of these marks in genes belonging to the same network indicates the DNA methylation involvement in a specific regulation pathway. For instance, in the Chinese/Japanese plum (*Prunus mume*)*,* many genes belonging to eight floral scent biosynthesis pathways show different methylation levels during the development of the flower [115]. These genes encode enzymes that are key regulators of floral scent production, such as the Coniferyl Alcohol Acetyltransferase (PmCFAT1a/1c) and the Benzyl Alcohol Acetyltransferase (PmBEAT36/3). The CFAT proteins belong to the acyltransferase family and are fundamental for eugenol synthesis, catalyzing the transformation of the substrate coniferyl alcohol to coniferyl acetate [116]. The BEAT proteins catalyze the transfer of the acetyl group of acetyl-CoA to the carbonyl group of benzyl alcohol, generating the benzyl acetate [117]. Whole-genome bisulfite sequencing in *P. mume* revealed that most of the differentially methylated genes are involved in different steps of the phenylpropane biosynthesis pathway through which over 90% of the floral volatiles are produced [115].

The dynamic of DNA methylation can vary in space and time, e.g., among different tissues or stages of development. The pineapple (*Ananas comosus*) leaves are an excellent example of the variation of DNA methylation during the diel (24 h). In the pineapple, the white leaf base is non-photosynthetic tissue. Here, the number of methylated genomic sequences is appreciably reduced across diel time course compared to the photosynthetic tissues, constituted by the green leaf tip. This alteration is particularly evident in the CHH sequences rather than in the CG and CGH context, where significant global differences are absent and occur in both gene and transposon regions. Temporal methylation data reveal that DNA methylation levels are low in green and white leaf tissue in the early morning and then increase during the afternoon (starting at 4 pm). In particular, a differentially methylated profile between green and white leaf tissues during the diel periods is characteristic of the CAM pathway genes, e.g., beta carbonic anhydrase (*beta-CA*), phosphoenolpyruvate carboxylase (*PEPC*), phosphoenolpyruvate carboxylase kinase (*PPCK*), and malate dehydrogenase (*MDH*). Therefore, DNA methylation can affect CAM photosynthesis in pineapple [118].

## 4. Methylation, Environment, and Evolution

Plants can respond to environmental stimuli by changing their epigenetic landscape. As proved by methylome analysis of different plant species, biotic or abiotic stress can affect DNA methylation, modifying the expression and regulation of stress-responsive genes [17].

### 4.1. Abiotic and Biotic Stimuli

In the orchid *Dendrobium officinale*, the cold and drought stress can influence the expression of the methylase *DoC5-MTase* and demethylase *DodMTase* genes. The promoter of these genes contains hormone-, light-, and stress-responsive *cis*-acting elements, confirming the relationship between the methylase/demethylase expression and the environmental conditions. In particular, the expression of most *DoC5MTases* is reduced under cold stress while the expression of *DodMTases* increases. Moreover, the *DodMTase* and *DoC5-MTase* gene transcription levels are correlated with the biosynthesis of water-soluble polysaccharides (WSPs). In fact, during the transition from protocorm-like bodies to plantlets in *Dendrobium*, the WSP content increases, and there is an up-regulation of both the *DodMTase* genes and the genes involved in WSP biosynthesis. At the same time, the expression of *DoC5-MTase* decreases. Environmental stresses, such as the cold, can affect the activity of methylase/demethylase transcription. This alteration reflects in a variation of the WSP synthesis and accumulation through regulating the genes involved in this pathway [119] (Table 1).

Similarly, in the tea plant (*Camellia sinensis*), the expression of the *cytosine-5 DNA methyltransferase* (*C5-MTase*) and *DNA demethylase* (*dMTase*) genes is dependent on environmental stimuli. As in *Dendrobium,* the promoters of *CsC5-MTase* and *CsdMTase* have multiple *cis*-acting elements responsive to light, phytohormones, and stress, suggesting that the relationship between environmental stimuli and DNA methylation is conserved among plants. The expression of the *Camellia* methylases and demethylases depends on abiotic stress, such as cold and drought. In particular, due to abiotic stress, the transcription of most *CsC5-MTases * is repressed, and that of all four *CsdMTase* genes increases [120], showing that the DNA methylation/demethylation activity can be a quick and effective response to environmental changes (Table 1).

There are many other examples where abiotic stress is a critical factor contributing to the modulation of DNA methylation. Cold is a stress factor that controls methylome flexibility in many plant species. *Arabidopsis* plants grown at two different temperatures that affect flowering, 10 °C and 16 °C, have a significant difference in the methylation in the CHH context. On the contrary, there are no considerable alterations in CG and CHG contexts. The differentially methylated regions (DMR) of the *Arabidopsis* plants subjected to the two different temperatures are located in transposable elements, according to their association with CHH methylation [121]. Active demethylation is due to low temperature in *Z. mays,* especially in root tissue [122,123] and in the snapdragon (*Antirrhinum*
*majus*), where it causes the activation of the Tam3 transposons [123,124,125].

In addition to temperature, other abiotic factors can affect DNA methylation. For example, salt stress induces demethylation in rape (*Brassica napus*) and *O. sativa* [123,126,127,128]. In *B. napus*, two different cultivars have distinct DNA methylation levels under salt stress, resulting in the variation of the expression of two stress-related genes: *Lacerata *(*LCR*) and *Trehalose Phosphatase*/Synthase 4 (*TPS4*).

The *LCR* gene encodes a cytochrome P450 monooxygenase involved in the cutin synthesis that controls the water loss. The *LCR* expression decreases in the salinity-tolerant Exagone cultivar due to increased methylation under high salt conditions. On the contrary, the *LCR* expression remains unaltered in the salinity-sensitive Toccata cultivar. The *TPS4* gene belongs to the biosynthetic pathway of trehalose, a disaccharide involved in response to dehydration. Under salinity stress, the expression of *TPS4 * increases in Exagone and decreases in Toccata, reflecting its methylation pattern [126] (Table 1).

A diversity of rice varieties respond to salt stress in different ways through DNA methylation changes. For example, under salt stress, the salt-tolerant Pokkali variety promptly removes its DNA methylation marks in comparison to the salt-sensitive variety IR29 [128].

As mentioned before, water deficit is an abiotic stress that affects DNA methylation, for example increasing the demethylation levels in tomato (*Solanum lycopersicum*) where the drought-responsive gene *Asr2* is hypomethylated in roots under drought conditions [123,129] (Table 1).

Salinity and drought stress also affect *Z. mays* plants, altering cell cycle regulation and chromatin remodeling. In particular, due to the exposure to this abiotic stress, DNA methylation levels are reduced together with different changes in histone modifications [130].

Arsenic toxicity is another abiotic stress affecting DNA methylation, as in the fern *Pteris cretica*, where arsenic stress reduces DNA methylation in old fronds. Changes in the methylation landscape directed by arsenic exposure alter several physiological parameters of this fern species and reduce the metabolite and water transport between roots and fronds [131].

The heavy metal stress also induces specific DNA methylation changes in the green alga *Scenedesmus acutus.* Chromium is one of the environment’s most diffused and toxic metals. The methylation pattern of two *S. acutus* strains with different chromium sensitivity is very different, suggesting that DNA methylation is involved in chromium tolerance in algae [132].

A further interesting example of the abiotic stress effect on methylation is the effective hypermethylation in the *Pinus silvestris* genome as a protective mechanism and adaptation strategy under the mutagenic ionizing radiations in the Chernobyl area [123,133].

In addition to abiotic factors, biotic stimuli such as the colonization of microorganisms or pathogens can influence DNA methylation levels. For example, infection of nematode cysts stimulates hypomethylation in soybean (*Glycine max*) and *A. thaliana* roots [134,135]. Many findings display that the DME mutants of *Arabidopsis* have an increased susceptibility to infection by pathogenic bacteria and fungi [136], suggesting that the regulation of DNA methylation is related to defense mechanisms against different biotic external agents.

### 4.2. Environmental Adaptations and Evolution

When environmental stimuli influence the reprogramming of DNA methylation in the germline, these modifications can be transmitted to the next generation [42]. In contrast to animals, where the germline is determined early during development, plants establish the germline later due to environmental signaling and cellular context.

The lack of an early separation between somatic and germline tissues in plants makes transgenerational transmission of epigenetic modifications more probable and straightforward [137,138]. The transgenerational epigenetic inheritance (TEI) in plants is frequently determined by environmental stimuli inducing epimutations that can be inherited across generations [139]. For example, some plants, such as *Arabidopsis*, acquire a specific methylation profile in response to drought stress. This methylation pattern can be transmitted to the next generations of stressed and unstressed plants. However, a prolonged irrigation period can cause the loss of the epialleles and related phenotype [140]. This suggests low stability of the epimutations, which tends to disappear over time and in the absence of the environmental stimulus.

Despite their instability, variations in DNA methylation and the consequent modification of gene expression can induce phenotypic changes that are evolutionarily advantageous. Natural selection can act on these epimutations, spreading or fixing epialleles that could become adaptive. According to this view, the external environment could contribute to plant evolution [141,142].

An exciting example of the species-specific methylation patterns resulting from the divergent selection driven by eco-environmental variables, such as water availability and temperature, is represented by three closely related allotetraploid orchid species: *Dactylorhiza majalis*, *D. traunsteineri*, and *D. ebudensis*. These orchid species originated from the same diploid parental lineages, *D. fuchsii* and *D. incarnata*; however, they appeared at different times during the late Quaternary. Their epigenetic variation, triggered by the doubling of the genome, reflects their evolutionary history, geographical distribution, and ecological niche. In particular, *D. majalis * shows high resistance to the humidity of the meadows in Western and Central Europe, the Baltic region, and northwestern Russia. *D. traunsteineri*, widespread in northwestern and central Europe, developed the capability to grow in the marshes. *D. ebudensis* is distributed principally in northwest Scotland and has poor moisture tolerance. Analyzing the epigenome of the three species, different epiloci related to vapor pressure and temperature show a species-specific methylation profile related to their ecology and evolution [143].

Different methylation patterns can also occur in ecotypes of the same species showing phenotypic divergence due to different environmental conditions. For example, tropical and temperate lotus (*Nelumbo nucifera*) are adapted to low and high latitudes, respectively, and present different methylation profiles. The rhizome methylome of the temperate lotus is hypermethylated compared to that of the tropical lotus, resulting in the differential expression of many genes related to the differentiation of the root tissue, such as starch-biosynthesis, gibberellin, and brassinosteroid-signaling genes. Due to this different methylation landscape and differential gene expression, the rhizome morphology shows evident differences between the two ecotypes, better adapted to the distinct ecological conditions: the tropical ecotype has thin rhizomes compared to the temperate, characterized by swollen roots [144].

Therefore, epimutations could be considered a significant evolutionary force for plants, and the mutant phenotypes obtained by methylation changes are rapidly exposed to the action of natural selection. In addition, methylated DNA regions have a high mutation rate, and they could represent the first step towards a more stable sequence mutation [138].

The partial reversibility of epimutations generates a phenotypic effect more dynamic than DNA sequence changes. One well-known example of gene silencing affecting plant morphology is the methylation of the *CYCLOIDEA* gene (*CYC*) in the common toadflax (*Linaria vulgaris*). In many angiosperms, the *CYC* gene has a conserved role in establishing bilateral symmetry, promoting the development of the dorsal identity of the flower organs [145]. During the angiosperm evolution, many flower symmetry transitions have occurred [146]. These reversions depend on the modifications of expression profile or mutations in the genes involved in determining flower symmetry. The loss or the reduced expression of the *CYC* gene in many angiosperms, such as *Conandron ramondioides* [147] or *Plantago lanceolata* [148], is associated with the ventralization of the flower [145]. In *L. vulgaris,* the *CYC* methylation, and the consequent reduction of its expression, causes the loss of bilateral symmetry, and the flower assumes a peloric mutant phenotype with radial symmetry [149]. The genetic knock-down by hypermethylation can function as the first step of adaptation before the complete loss of the gene function. It guarantees the possibility of implementing a co-adaptation process of other genes that can compensate for the gene loss by generating a functional balance [42,138].

Methylation levels vary among plant species, and this also depends on the genome content of repetitive and mobile DNA, which has high methylation levels [40]. Therefore, DNA methylation is an evolutionary trait related to plant genome size and complexity. However, the relationship between global cytosine methylation and genome size is not linear and is more complex than a simple cause-effect relation [150].

## 5. Conclusions

DNA methylation has acquired many functions during plant evolution (Figure 3). The ubiquitous presence in all plant species of the methylation of transposable elements demonstrates its very ancient role. The regulation of gene expression is another central function of DNA methylation. This regulatory process is present in all land plants, as well as the methylation of the coding regions. In angiosperms, there is an expansion of the role of methylation. The intervention of PolV in the RdDM pathway could have played a role in reducing genome size and enhancing the diploidization after whole genome duplications [42,151]. Likewise, the maintenance of methylation of non-CG sequences by CMT3 is another angiosperm-specific regulatory mechanism. Finally, genomic imprinting represents a fundamental evolutionary novelty in flowering plants [42].

DNA methylation is one of the best-known epigenetic mechanisms and plays a central role in plant development and evolution. During the plant life cycle, gene expression, transposon mobility, and chromosome interactions depend on the balance between DNA methylation and demethylation. In addition, the environmental stimuli can affect the methylation landscape, resulting in specific adaptation and evolution of traits. However, fully understanding all the functions and mechanisms of mRNA and DNA methylation and its repercussions on the evolution of plant species is still a fascinating challenge.

Up to now, crop improvement has been based on the analysis of genetic variation and the selection of advantageous alleles. In addition to this “classical” approach, the knowledge of epigenetic mechanisms can be applied to develop “epimutagenesis” techniques to modify and improve characters of agronomic interest [34].

In human cells, the application of synthetic epigenetic regulation to control gene expression is already used [79,152]. To date, technologies are being developed to enhance agriculture, manage diseases, and improve crops. One of the most recent strategies is using exogenous dsRNA to induce gene silencing and epigenetic alterations through RNA interference (RNAi) in plants [153].

Currently, advances are being made for locus-specific manipulation of DNA methylation. One of the most exciting approaches is the use of proteins that bind DNA (e.g., zinc finger proteins, dCas9 proteins) fused with RdDM components and TET-like catalytic domains to direct locus-specific DNA demethylation in plant genomes and modify economically and agriculturally significant plant traits [154].

In conclusion, new knowledge of different epigenetic mechanisms will help develop genetic engineering strategies to obtain a synthetic epigenetic regulation that can be an essential tool for plant breeding [79].

## Figures and Tables

**Figure 1 ijms-23-08299-f001:**
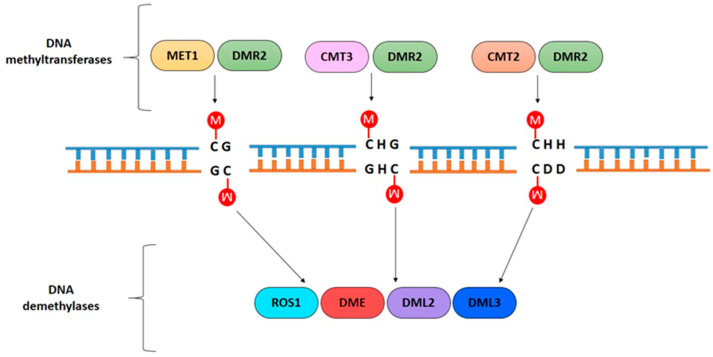
Specific DNA methyltransferases and demethylases mediate cytosine methylation (red circle) in different sequence contexts. CG, CHG, and CHH methylation are carried out by MET1, CMT3, and CMT2, respectively. DRM2, involved in the RdDM pathway, regulates all sequence context methylation. ROS1, DME, DML2, and DML3 act as demethylases.

**Figure 2 ijms-23-08299-f002:**
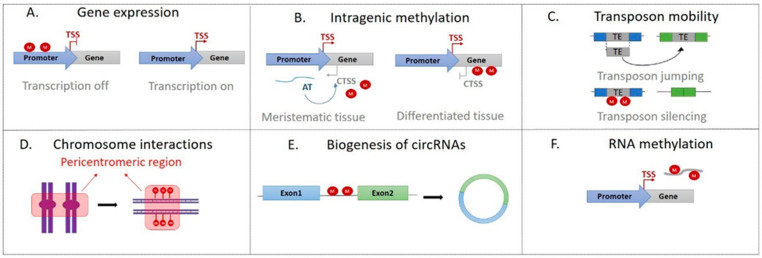
Different roles of DNA methylation in plant cells. (**A**) The promoter DNA methylation (red circle) represses transcription activity and gene expression; (**B**) methylation of the coding regions generates an inaccessible chromatin structure suppressing the aberrant transcription start site; (**C**) DNA methylation affects genome stability by silencing transposons and other DNA repeated sequences; (**D**) chromosome interactions through pericentromeric regions or heterochromatin islands depend on the methylation of these regions; (**E**) DNA methylation could be involved in the biogenesis of circRNAs; (**F**) mRNA methylation controls stability, splicing, and processing of the transcript itself. TSS, transcription start site; CTSS, cryptic transcription start site; AT, aberrant transcript; TE, transposable elements.

**Figure 3 ijms-23-08299-f003:**
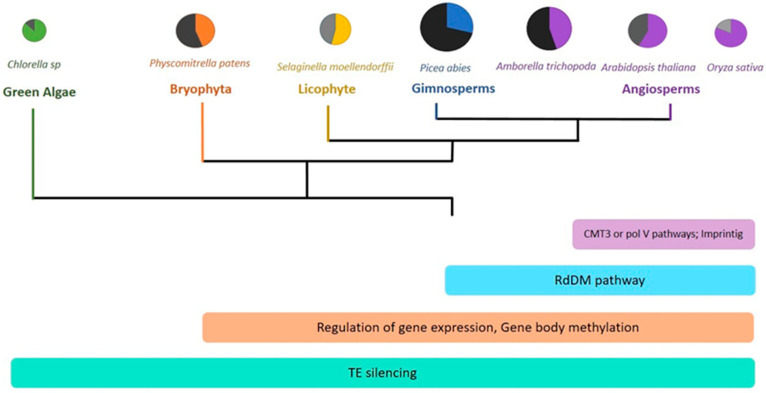
DNA methylation during plant evolution. The diameter of the colored circles represents the genome size of selected species; the dark slice represents the percentage of repeated sequences, and the intensity of the gray represents the level of global DNA methylation. Across plant evolution, DNA methylation assumes new roles represented by colored rectangles. The TE silencing constitutes the most conserved function of methylation. More recently, novel regulation pathways have emerged.

**Table 1 ijms-23-08299-t001:** Summary of the genes that change the methylation levels in response to abiotic stress.

Species	Genes	Protein	Phenotype	Stress	Methylation State
*D. officinale*	*DoC5-MTase*	methylase	biosynthesis of WSPs	cold	methylated
*D. officinale*	*DodMTase*	demethylase	biosynthesis of WSPs	cold	demethylated
*C. sinensis*	*C5-MTase*	methylase		cold/drought	methylated
*C. sinensis*	*CsdMTase*	demethylase		cold/drought	demethylated
*B. napus* Exagone	*LCR*	cytochrome P450 monooxygenase	cutin synthesis	salt	methylated
*B. napus* Toccata	*LCR*	cytochrome P450 monooxygenase	cutin synthesis	salt	unaltered
*B. napus* Exagone	*TPS4*	trehalose phosphatase/synthase 4	biosynthetic pathway of trehalose	salt	demethylated
*B. napus* Toccata	*TPS4*	trehalose phosphatase/synthase 4	biosynthetic pathway of trehalose	salt	methylated
*S. lycopersicum*	*Asr2*	putative transcription factor	response to water-stress	drought	demethylated

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
