# Peer review of "Plant DNA Methylation: An Epigenetic Mark in Development, Environmental Interactions, and Evolution"

_ijms, 2022, doi:10.3390/ijms23158299_

Round 1
Reviewer 1 Report
Plant DNA methylation: an epigenetic mark in development, environmental interactions, and evolution
General comments: This manuscript aims to review updated knowledge regarding the DNA methylation flexibility and role in plant cells and development, in responses to environmental changes and in evolution. It is an important contribute for knowledge dissemination. It ends up with a perspective regarding challenges for its application for achieving specific plant traits with meaning for plant breeding purposes. The all MS is very well written and organized containing literature landmarks that covers crucial questions regarding the role of epigenetics in regulation of gene expression and plant traits.
Specific comments:
Lines 66-72- You could complete including a reference to the concept methylation/demethylation cycle can be more updated including for example the role of active DNA demethylation with the involvement of many enzymes. No reference to the role of TET enzymes in the dynamics of all this process.
Lines 101-104- This paragraph seems confuse and in way may be contradictory to what is written in lines 136-137. Please, check and clarify the message.
Lines 395-98- This paragraph could be expanded in order to include the concept of what is considered to be a transgenerational inheritance of epigenetic marks (e.g. how many generations are needed).
Author Response
Reviewer 1
General comments: This manuscript aims to review updated knowledge regarding the DNA methylation flexibility and role in plant cells and development, in responses to environmental changes and in evolution. It is an important contribute for knowledge dissemination. It ends up with a perspective regarding challenges for its application for achieving specific plant traits with meaning for plant breeding purposes. The all MS is very well written and organized containing literature landmarks that covers crucial questions regarding the role of epigenetics in regulation of gene expression and plant traits.
Specific comments:
Lines 66-72- You could complete including a reference to the concept methylation/demethylation cycle can be more updated including for example the role of active DNA demethylation with the involvement of many enzymes. No reference to the role of TET enzymes in the dynamics of all this process.
Answer: We added a reference to the methylation/demethylation cycle and the TET enzymes, also including three new references:
- Wu, X. and Y. Zhang, TET-mediated active DNA demethylation: mechanism, function and beyond. Nat Rev Genet, 2017. 18(9): p. 517-534.
- Mahmood, A.M. and J.M. Dunwell, Evidence for novel epigenetic marks within plants. AIMS Genet, 2019. 6(4): p. 70-87.
- Ji, L., et al., TET-mediated epimutagenesis of the Arabidopsis thaliana methylome. Nat Commun, 2018. 9(1): p. 895.
Lines 101-104- This paragraph seems confuse and in way may be contradictory to what is written in lines 136-137. Please, check and clarify the message.
Answer: We checked and clarified this paragraph (101-104) and slightly modified the lines 177-178.
Lines 395-98- This paragraph could be expanded in order to include the concept of what is considered to be a transgenerational inheritance of epigenetic marks (e.g. how many generations are needed).
Answer: We expanded this paragraph, including the concept of transgenerational epigenetic inheritance, also including two new references:
- Paszkowski, J. and U. Grossniklaus, Selected aspects of transgenerational epigenetic inheritance and resetting in plants. Curr Opin Plant Biol, 2011. 14(2): p. 195-203.
- Sadhukhan A , P.S., Mitra J , Siddiqui N , Sahoo L, Kobayashi Y , Koyama H, How do plants remember drought? Planta, 2022. 256.
Reviewer 2 Report
1- The authors have to recreate most of the figures so we can get more information (figures have to be collective) and easily understand the figure so I would recommend that they prepare the figures in a more professional way (not so simple and have less information or so complicated and crowded)
2- I would recommend including collective tables indicating genes organized, their ontology, feedback of organization, impact on the phenotype............etc
3- Language, grammar, and punctuation require to be checked
4- I would recommend citing this reference
Kamal, K. Y., Khodaeiaminjan, M., Yahya, G., El-Tantawy, A. A., Abdel El-Moneim, D., El-Esawi, M. A., Abd-Elaziz, M., & Nassrallah, A. A. (2021). Modulation of cell cycle progression and chromatin dynamic as tolerance mechanisms to salinity and drought stress in maize. Physiologia plantarum, 172(2), 684–695. https://doi.org/10.1111/ppl.13260
Author Response
Reviewer 2
1- The authors have to recreate most of the figures so we can get more information (figures have to be collective) and easily understand the figure so I would recommend that they prepare the figures in a more professional way (not so simple and have less information or so complicated and crowded)
Answer: We checked and modified Figures 1 and 3. In addition, we changed Figure 2.
2- I would recommend including collective tables indicating genes organized, their ontology, feedback of organization, impact on the phenotype............etc
Answer: We added Table 1 with the summary information on genes, phenotype, methylation profile, and kind of stress for different plant species (cited in the text).
3- Language, grammar, and punctuation require to be checked
Answer: We checked language, grammar, and punctuation.
4- I would recommend citing this reference
Kamal, K. Y., Khodaeiaminjan, M., Yahya, G., El-Tantawy, A. A., Abdel El-Moneim, D., El-Esawi, M. A., Abd-Elaziz, M., & Nassrallah, A. A. (2021). Modulation of cell cycle progression and chromatin dynamic as tolerance mechanisms to salinity and drought stress in maize. Physiologia plantarum, 172(2), 684–695. https://doi.org/10.1111/ppl.13260
Answer: We added this reference and cited it within the text.
Reviewer 3 Report
This rewiev article introduces the significance of the DNA methylation as an epigenetic mark through the plant ontogenesis, development, evolution and environmental adaptations.
I find the approach to the topic interesting and novel. I would like to highlight that this review manuscript is basically well written, well understood and easy to follow. The number of citations is sufficient, but a little more could refer to new(er) article containing the results of the last 5 years (2018-2022). This currently account for about 19% of the manuscript. However, I think that what the article presents about the importance of DNA methylation in taxa under angiosperms is not enough. In connection with these, it would be worthwhile to mention some more species and results.
Conclusions:
This section is Ok, but I think, the practical use of DNA methylation could also be introduced here, even with interesting ideas and methods. This review article points out that DNA methylation can affect the development of a plant and its response to stressors at many points therefore it can be an effective tool to protect the yield of crop plants. This may even affect genetic modification and non-genetic techniques for the feasibility of sustainable crop protection with DNA methylation as well. These should also be added to the Abstract.
I would like to draw the attention of the authors to some minor errors:
The figures must be inserted in the appropriate place (after the first citationss in the main text cf.: instructions for authors/preparing figs, schemes and tables).
Line 228-246: Note that this type of embryo sac or megagametophyte is only one among many others. What you mentioned, is Polygonum type (and it cannot even be said that it is the most common), there are exist Allium-, Oenothera-, Adoxa type, ect. Were the others also examined for DNA methylation? If not, this can also be highlighted in the conclusion section.
Please romove the highlighting the species names (L. 347-356) and titles of the citations (in the References).
Write the common English name of each species (where applicable, exist), and write their scientific name in parentheses after it. After that, the short Latin name of the species can then be used. Please chech it throught the MS. Such as L. 301: Chinese/Japanese plum (Prunus mume) … L. 309: P. mume..
L. 390 : Please indicate the Latin name of soybean (Glycine max).
L. 422: Please correct the Latin name of lotus and put it in brackets! …lotus (Nelumbo nucifera)..
Author Response
Reviewer 3
This rewiev article introduces the significance of the DNA methylation as an epigenetic mark through the plant ontogenesis, development, evolution and environmental adaptations.
I find the approach to the topic interesting and novel. I would like to highlight that this review manuscript is basically well written, well understood and easy to follow. The number of citations is sufficient, but a little more could refer to new(er) article containing the results of the last 5 years (2018-2022). This currently account for about 19% of the manuscript. However, I think that what the article presents about the importance of DNA methylation in taxa under angiosperms is not enough. In connection with these, it would be worthwhile to mention some more species and results.
Answer: We added new, more recent references:
- Wu, X. and Y. Zhang, TET-mediated active DNA demethylation: mechanism, function and beyond. Nat Rev Genet, 2017. 18(9): p. 517-534.
- Mahmood, A.M. and J.M. Dunwell, Evidence for novel epigenetic marks within plants. AIMS Genet, 2019. 6(4): p. 70-87.
- Ji, L., et al., TET-mediated epimutagenesis of the Arabidopsis thaliana methylome. Nat Commun, 2018. 9(1): p. 895.
- Sadhukhan A , P.S., Mitra J , Siddiqui N , Sahoo L, Kobayashi Y , Koyama H, How do plants remember drought? Planta, 2022. 256.
- Kamal, K. Y., Khodaeiaminjan, M., Yahya, G., El-Tantawy, A. A., Abdel El-Moneim, D., El-Esawi, M. A., Abd-Elaziz, M., & Nassrallah, A. A. (2021). Modulation of cell cycle progression and chromatin dynamic as tolerance mechanisms to salinity and drought stress in maize. Physiologia plantarum, 172(2), 684–695. https://doi.org/10.1111/ppl.13260
- Han ML, Y.J., Zhao YH, Sun XW, J Meng JX, Zhou J, Shen T, Li HH , Zhang F, How the Color Fades From Malus halliana Flowers: Transcriptome Sequencing and DNA Methylation Analysis. Front Plant Sci, 2020. 11.
- Zemanova, V., et al., Effect of arsenic stress on 5-methylcytosine, photosynthetic parameters and nutrient content in arsenic hyperaccumulator Pteris cretica (L.) var. Albo-lineata. BMC Plant Biol, 2020. 20(1): p. 130.
- Ferrari, M., et al., Role of DNA methylation in the chromium tolerance of Scenedesmus acutus (Chlorophyceae) and its impact on the sulfate pathway regulation. Plant Sci, 2020. 301: p. 110680.
- Kordyum and Mosyakin 2020, Endosperm of Angiosperms and Genomic Imprinting. Life (Basel) 10(7): 104.
We added more species and more examples, e.g., Malus halliana, Pteris cretica, Scenedesmus acutus.
Conclusions: This section is Ok, but I think, the practical use of DNA methylation could also be introduced here, even with interesting ideas and methods. This review article points out that DNA methylation can affect the development of a plant and its response to stressors at many points therefore it can be an effective tool to protect the yield of crop plants. This may even affect genetic modification and non-genetic techniques for the feasibility of sustainable crop protection with DNA methylation as well. These should also be added to the Abstract.
Answer: We introduced the possible applications of DNA methylation knowledge in the Conclusion section and added a sentence in the abstract.
I would like to draw the attention of the authors to some minor errors:
- The figures must be inserted in the appropriate place (after the first citationss in the main text cf.: instructions for authors/preparing figs, schemes and tables).
Answer: We moved the figures to the appropriate place.
Line 228-246: Note that this type of embryo sac or megagametophyte is only one among many others. What you mentioned, is Polygonum type (and it cannot even be said that it is the most common), there are exist Allium-, Oenothera-, Adoxa type, ect. Were the others also examined for DNA methylation? If not, this can also be highlighted in the conclusion section.
Answer: We specified that we were referring to the Polygonum-type embryo sac and that, to date, genomic imprinting is described only in species with this type of embryo sac. We added two new references:
- Kordyum and Mosyakin 2020, Endosperm of Angiosperms and Genomic Imprinting. Life (Basel) 10(7): 104.
- Rudall 2006 How many nuclei make an embryosac in flow-ering plants? BioEssays 28:1067–1071.
- Please romove the highlighting the species names (L. 347-356) and titles of the citations (in the References).
Answer: We corrected the citation style.
- Write the common English name of each species (where applicable, exist), and write their scientific name in parentheses after it. After that, the short Latin name of the species can then be used. Please chech it throught the MS. Such as L. 301: Chinese/Japanese plum (Prunus mume) … L. 309: P. mume..
- 390 : Please indicate the Latin name of soybean (Glycine max).
- 422: Please correct the Latin name of lotus and put it in brackets! …lotus (Nelumbo nucifera)..
Answer: We checked ad corrected throughout the MS the common/Latin names of the species.
Round 2
Reviewer 2 Report
I think the authors improved a lot there manuscript, I would accept the manuscript in the present form for publication
Author Response
Thank you very much.